# Color Changes in Ag Nanoparticle Aggregates Placed in Various Environments: Their Application to Air Monitoring

**DOI:** 10.3390/nano11030701

**Published:** 2021-03-11

**Authors:** Koichi Ozaki, Fumitaka Nishiyama, Katsumi Takahiro

**Affiliations:** 1Faculty of Materials Science and Engineering, Kyoto Institute of Technology, Matsugasaki, Sakyo, Kyoto 606-8585, Japan; ozaki2008@gmail.com; 2Research Institute for Nanodevice and Bio Systems, Hiroshima University, Kagamiyama 1-4-1, Higashi-Hiroshima, Hiroshima 739-8527, Japan; fnishi@hiroshima-u.ac.jp

**Keywords:** Ag nanoparticle, localized surface plasmon resonance, color, red-shift, blue-shift

## Abstract

Fresh Ag nanoparticles (NPs) dispersed on a transparent SiO_2_ exhibit an intense optical extinction band originating in localized surface plasmon resonance (LSPR) in the visible range. The intensity of the LSPR band weakened when the Ag NPs was stored in ambient air for two weeks. The rate of the weakening and the LSPR wavelength shift, corresponding to visual chromatic changes, strongly depended on the environment in which Ag NPs were set. The origin of a chromatic change was discussed along with both compositional and morphological changes. In one case, bluish coloring followed by a prompt discoloring was observed for Ag NPs placed near the ventilation fan in our laboratory, resulted from adsorption of large amounts of S and Cl on Ag NP surfaces as well as particle coarsening. Such color changes deduce the presence of significant amounts of S and Cl in the environment. In another case, a remarkable blue-shift of the LSPR band was observed for the Ag NPs stored in the desiccator made of stainless steel, originated in the formation of CN and/or HCN compounds and surface roughening. Their color changed from maroon to reddish, suggesting that such molecules were present inside the desiccator.

## 1. Introduction

Environmental pollution has become an important problem all over the world. For instance, nitrogen oxide (NO_x_), which plays a major role in the formation of ozone and acid rain, is one of the most dangerous air pollutants. Continued exposures to NO_2_ cause increased incidence of acute respiratory infection [1,2]. In addition, sulphur compounds such as sulphur oxides (SO_x_) and hydrogen sulphide (H_2_S) are also well known to air pollutants. Even short-term exposures to SO_2_ are linked with respiratory effects including breathing difficulty and asthma symptoms [1,2]. The H_2_S, produced by many industrial processes and decomposition of oil, is a very poisonous, corrosive, flammable and explosive gas [3].

Focusing on industrial materials, e.g., electronics and semiconductor material, copper and silver are extensively used because of their high electrical conductivity, ductility and malleability. They are, however, inevitably corroded by reacting with H_2_S in ambient air to produce their sulfide such as Cu_2_S and Ag_2_S [4,5,6]. The atmospheric corrosion caused by the pollutant gases is becoming a significant factor in the reliability of electrical equipment. As electronics continue to decrease in size, it is important to be aware of the pollutant gases around the electrical equipment to prevent corrosion risk. 

In recent years, particulate matter (PM) has also attracted considerable attention due to adverse effects to human health and materials corrosion. It is known to that the PM can be carried deep into the lungs, worsening lung diseases [1]. Moreover, adsorption of PM containing sulphur compounds on copper and its alloys corrodes locally the metals, so-called “pitting corrosion”, which causes degradation of their performance. Therefore, the metal corrosion by the PM would be a serious problem [7]. 

The detecting or monitoring of both gaseous pollutants and PM is required to preserve human health and safety. The detection of gaseous pollutants is, for instance, typically achieved using a semiconductor metal oxide [8,9], and electrochemical [10,11] and optical sensors [12,13]. Although the semiconductor metal oxide sensors have advantages of a fast response time, they always require an electric power, making long-term personal and mobile monitoring difficult [14]. The electrochemical sensors have also good sensitivity and short response time, but their usage is limited by temperature and humidity [15]. Therefore, they are unsuitable for use in many harsh environments. On the other hand, optical sensors are essentially unaffected by the temperature and humidity. In particular, colorimetric sensors have attracted an interest [16,17,18,19] because they can minimize efforts associated with instrumentation and operation in detection, making them easily applicable to on-site detection [20,21]. The sensors mentioned above are preferable for prompt detection of gaseous pollutants. 

It is sometimes necessary to monitor gaseous pollutants and PM accumulated in a human body and electronics materials to realize sustainable life. In this case, long-term monitoring, rather than quick sensing, of such harmful substances is needed. This situation is analogous to the monitoring of cumulative radiation dose using the film badge dosimeter for radiation protection. In this paper, we propose that a silver nanoparticle (Ag NP) aggregate deposited on a transparent substrate can be candidate for long-term monitoring of the cleanliness of a living or working atmosphere. The reason why elemental Ag is suitable for such monitoring originated from the high reactivity of Ag with harmful species including N_2_O and CO_2_ [22]. Furthermore, metallic Ag particles in several tens of nanometers have a strong optical extinction band originating in localized surface plasmon resonance (LSPR) in the visible range. Both the optical extinction and wavelength of LSPR, i.e., Ag NP color, strongly depend on the composition and thickness of a contaminant layer formed on Ag NP surfaces by reacting with harmful species [23,24]. Morphological alterations in an Ag NP aggregate occur after the Ag_2_S formation [25], which also change LSPR characteristics. Thus, chemical changes as well as morphological alterations in the Ag NP aggregate, when reacted with harmful species, lead to its color changing, enabling one to monitor the cleanness of atmosphere in which Ag NPs are placed. In fact, our previous study [26] showed that the color of an Ag NP aggregate stored in ambient laboratory air for 46 days greatly changed along with compositional and morphological changes. A chromatic change is easily recognized when transparent materials such as SiO_2_ are used as a substrate for deposition of a Ag NP aggregate.

Our final goal is to develop a small and cheap device to check the cleanness of atmosphere by monitoring chromatic changes in Ag NPs/SiO_2_. To realize this, we need to acquire the basic knowledge of LSPR characteristics of a Ag NP aggregate placed in various environments for several weeks. The LSPR characteristics are expected to be different from place to place because components of the atmosphere depend on storage environments of Ag NPs/SiO_2_. In the present work, firstly optical extinction spectroscopy of Ag NPs/SiO_2_ samples is conducted after they are placed in various environments for several weeks. Then the origin of the spectral changes, i.e., chromatic changes, is examined from the viewpoint of compositional and morphological changes in Ag NPs. Finally, the cleanliness of the atmosphere in which Ag NPs are placed is discussed by the chromatic changes to develop an environment monitoring device in future.

## 2. Materials and Methods

The direct current (dc) sputtering method was used to fabricate Ag NPs on four kinds of substrates including fused silica (SiO_2_) plates (T-2630, CoorsTek KK, Tokyo, Japan), single crystalline Si (100) wafers (Cz, n-type, Shin-Etsu Chemical Co. Ltd., Tokyo, Japan), highly oriented pyrolytic graphite (HOPG) plates (PGCX20, Panasonic Production Engineering Co., Ltd., Kadoma, Japan) and polyethylene naphthalate (PEN) foils (Teonex^®^, Teijin, Tokyo, Japan). These samples are referred to as Ag NPs/SiO_2_, Ag NPs/Si, Ag NPs/HOPG and Ag NPs/PEN, hereafter. A desktop rotary-pumped sputter coater, SC-701 MKII, developed by Sanyu Electron Co. Ltd., Tokyo, Japan was used with a silver sputtering target (purity: 99.8%). The pressure of the Ar gas was ~1 Pa, and the current was typically 7 mA at a voltage of 1.2 keV during the sputter deposition with a 30 s duration. The samples became gradually darker immediately after the deposition. Therefore, the samples were aged in vacuum (<7.0 × 10^−5^ Pa) for 7 days prior to air exposure, in order to stabilize the Ag NPs, e.g., morphology and crystallinity. The samples were stored at four different places; on a table (in ambient air), near a ventilation fan, inside a clean desiccator (class 100 of cleanroom classification) made of acrylic resin, inside a conventional desiccator (referred to as “metal desiccator”) made of stainless steel, in our laboratory where the illumination light was almost turned off. All the Ag NP samples of Ag NPs/SiO_2_, Ag NPs/Si, Ag NPs/HOPG and Ag NPs/PEN were subjected to the same conditions.

The Ag NPs/SiO_2_ samples were used for visual confirmation for chromatic changes as well as Ultraviolet–visible (UV–Vis) measurements. The color of the as-prepared Ag NPs/SiO_2_ sample (aged in vacuum for 7 days) was maroon as shown in Figure 1a. UV–Vis extinction spectra were taken in standard transmission geometry in the wavelength range of 200–800 nm.

The Ag NPs/Si samples were fabricated to observe morphology of Ag NP aggregates with atomic force microscopy (AFM). In the AFM observation, the dynamic force mode, corresponding to the tapping mode, of the Nanonavi Station manufactured by SII Nanotechnology Inc., Chiba, Japan was employed with a Si cantilever, SI-DF20, with a tip radius of curvature less than 10 nm. The lateral sizes of individual Ag NPs in the as-prepared Ag NPs/Si sample (aged in vacuum for 7 days) were 20–30 nm as shown in Figure 1b. The root mean square (RMS) surface roughness of the as-prepared Ag NPs/Si was estimated to be 2.1 nm.

The Ag NPs/HOPG samples were used to analyze chemical compositions and bonds at Ag NP surfaces using Rutherford backscattering spectrometry (RBS) and X-ray photoelectron spectroscopy (XPS). RBS was conducted with 2 MeV-^4^He ions produced from a single-ended Van de Graaff (VdG) accelerator of Hiroshima University. Backscattered ^4^He ions were detected with a surface barrier detector placed at an angle of 165° with respect to an incident beam. RBS analysis also can provide the areal density of Ag atoms in the samples. A typical areal density was estimated to be 1.6 × 10^16^ Ag atoms/cm^2^. XPS using Mg Kα radiation (*hν* = 1253.6 eV) was performed with a JEOL 9010 X-ray spectrometer (JEOL Ltd., Akishima, Japan).

The Ag NPs/PEN samples were prepared to detect heavy elements adsorbed on Ag NP surfaces using particle-induced X-ray emission (PIXE). The beams of 2 MeV-protons produced from the VdG accelerator and a Si (Li) detector with an appropriate filter made of polyethylene terephthalate was used for the PIXE analysis.

## 3. Results

Figure 2 shows the photographs of Ag NPs/SiO_2_ samples stored in various environments. All the Ag NPs/SiO_2_ samples were aged in vacuum for 7 days prior to air exposure. The color of the as-prepared Ag NPs/SiO_2_ sample was maroon as shown in Figure 1a and insets in each photograph in Figure 2. The color faded gradually for the sample stored in ambient air. The chromatic change from maroon to burgundy were clearly recognizable after 14 days. For the Ag NPs/SiO_2_ stored into the clean desiccator, its color turned still light violet even after 56 days. For the Ag NPs/SiO_2_ stored into the metal desiccator, the color became terracotta after 14 days and almost transparent after 56 days. The color of the Ag NPs/SiO_2_ stored near the ventilation fan faded quickly even after 14 days. This sample looked almost transparent after storing for 56 days.

The chromatic changes can be examined in a quantitative way using optical extinction spectroscopy. Figure 3 shows the optical extinction spectra in the wavelength range 200–800 nm obtained from the four kinds of Ag NPs/SiO_2_ samples presented in Figure 2. All the Ag NPs/SiO_2_ samples were aged in a vacuum for 7 days prior to air exposure. After the aging in a vacuum, the LSPR band intensity and wavelength of Ag NPs/SiO_2_ stored in ambient air were 505 ± 6 nm and 0.38 ± 0.02, respectively. The intensity of the LSPR band located at 400–700 nm gradually weakened with ambient air exposure. For the Ag/SiO_2_ stored in the clean desiccator, the rate of LSPR intensity decrease was lower than that for the sample stored in ambient air. The intensity of the LSPR band maintained at 0.16 even after 56 days. For the Ag NPs/SiO_2_ stored into the metal desiccator, the LSPR band shifted toward shorter wavelengths drastically. The most remarkable decrease in the LSPR band intensity was observed for the sample stored near the ventilation fan. The reproducibility of chromatic changes and optical extinction spectra of each Ag NPs/SiO_2_ sample was confirmed.

Figure 4a shows the RBS spectra acquired with 2 MeV-^4^He ions of as-prepared Ag NPs/HOPG aged in vacuum for 7 days. As shown in Figure 4a, a high-energy edge appears at approximately 0.51 MeV, corresponding to the scattering 2 MeV-^4^He ions from the C substrate. Even for the as-prepared Ag NPs/HOPG, small peaks corresponding to extrinsic contaminants of N (~0.63 MeV), O (~0.73 MeV) and S (~1.22 MeV) were detected. Figure 4b–e show the RBS spectra of Ag NPs/HOPG after air exposure for 14 days and 44 days. For the Ag NPs/HOPG stored in ambient air, into the clean desiccator and into the metal desiccator, N, O and S were detected. In addition to these elements, a small amount of Cl was detected on Ag NPs/HOPG stored in ambient air for 44 days. In the RBS spectra of Ag NPs/HOPG stored near the ventilation fan, peaks corresponding to contaminant elements such as N, O, S, Cl were found. In addition to assignable peaks for such contaminants, there are some peaks, for example, at energies of 1.64 MeV (in Figure 4c) and 1.85 MeV (in all the spectra), that cannot be determined by RBS because of its poor mass resolution for heavy elements. Therefore, we tried to detect trace amounts of heavy element impurities adsorbed on Ag NPs by PIXE analysis using proton beams.

Figure 5 shows the PIXE spectra of the Ag/PEN samples stored in various environments. Very weak peaks at 3.61 and 3.84 keV overlapped with background signals corresponding to Sb Lα_1_ and Lβ_1_ X-rays, respectively, which appear even for a bare PEN, indicating that Sb comes from the foil. Two intense peaks of 22.2 and 24.9 keV correspond to Ag Kα_1_ and Ag Kβ_1_ X-ray, respectively. Unfortunately, the impurity elements heavier than Ag, except for Sb, were not detected by the proton-PIXE. In Figure 5c, several peaks, e.g., Cr Kα_1_ (5.41 keV), Fe Kα_1_ (6.40 keV), Ni Kα_1_ (7.47 keV) X-rays, originated in extrinsic impurities adsorbed on Ag NPs. These impurities are coincident with the constituent elements of the metal desiccator. Furthermore, in Figure 5c,d, the Br Kα_1_ X-ray was observed at an energy of 11.92 keV, indicative of adsorption of Br on Ag NPs.

Figure 6a displays the narrow-scan XPS C 1s spectra of Ag NPs/HOPG after air exposure for 14 days. The C 1s binding energy (BE) for the Ag NPs/HOPG stored into the metal desiccator was higher than that for other samples, as can be seen in Figure 6a. For the sample stored into the metal desiccator, an intense N 1s peak was clearly observed as shown in Figure 6b. The higher C 1s BE and the intense N 1s peak indicate that CN and/or HCN compound forms on Ag NP surfaces [27]. Figure 6c shows the narrow-scan Ag 3d spectra of Ag NPs/HOPG. As demonstrated by the previous XPS study [28], the Ag 3d_5/2_ BE of 368.6 eV is assigned to Ag_2_S. Since the Ag 3d binding energies for a metallic Ag and Ag compounds such as AgCl, Ag_2_S, AgO and Ag_2_O are 367.9 ± 0.4 eV [29,30,31], it is difficult to assign each peak. In XPS analysis, the nature of Ag can be clearly seen in 4d valence band spectra as shown in Figure 6d. The valence band spectrum of the Ag NPs/HOPG stored into the metal desiccator became very sharp compared to that of the as-prepared Ag NPs/ HOPG. The band width, originating from spin-orbit splitting, for the as-prepared sample and Ag NPs/HOPG stored into the metal desiccator were 3.3 eV and 1.9 eV, respectively. In contrast, the spectral changes for the samples placed in ambient air, inside the clean desiccator and near the ventilation fan were rather small, but the band widths for these three samples were slightly narrower than that for the as-prepared sample. The observed band narrowing is evidence that Ag reacts with outermost layer of contaminants to partly form compounds such as AgCl, Ag_2_S and AgCN. The band width of the Ag NPs/HOPG stored in ambient air, into the clean desiccator and near the ventilation fan were 3.1 eV, 3.2 eV and 2.8 eV, respectively, after 14 days. Further band narrowing has been recognized in valence band spectra of Ag NPs samples stored for 56 days (not shown here).

Figure 7 shows the AFM images of the Ag NPs/Si sample stored in various environments for 14 days. Surface morphology of the Ag NPs samples after air exposure was different in roughness from that of the as-prepared Ag NPs. The particle coarsening as a result of coalescence was observed in all the samples. The RMS surface roughness of the Ag NPs/Si stored in ambient air, into the clean desiccator, into the metal desiccator and near the ventilation fan, were 3.6 nm, 3.5 nm, 6.2 nm and 4.4 nm, respectively. The most roughened surface was produced on the sample stored inside the metal desiccator, and for the sample stored in the clean desiccator the change in roughness was found to be smallest. In addition, worm-like aggregates of ~150 nm in length and ~50 nm in width were observed at the surface of the sample stored near the ventilation fan.

## 4. Discussion

The present work has shown that the adsorbed impurities and the color depends on the environments where Ag NPs/SiO_2_ were stored for a couple of weeks. This suggests that environmental substances have an effect on the change in color of Ag NPs/SiO_2_. Conversely, the chromatic change of Ag NPs/SiO_2_ is indicative of environmental substances, enabling one to check the quality or cleanliness of the environment. In this section, chromatic changes of Ag NPs/SiO_2_ are discussed for each sample in terms of compositional and morphological changes, and then the quality of environment in which Ag NPs/SiO_2_ were stored is also dealt with. For this purpose, chromatic changes measured by LSPR characteristics including resonant wavelength and intensity are represented in Figure 8. The LSPR characteristics of the four samples largely change in 14 days. In particular, the normalized LSPR intensity for the sample placed near the ventilation fan decreased down to 0.4 at elapsed time of 14 days as shown in Figure 8a. For this sample, the most drastic positive shift (+55 nm) was observed at that time as shown in Figure 8b. Visual compositional changes in Ag NPs within 14 days, obtained from Table 1, are also shown in Figure 9. The largest increase in impurity elements O (0.11 to 0.27), S (0.04 to 0.14) and Cl (0 to 0.06) are found in the atomic ratio for the sample placed near the ventilation fan. Next, chromatic changes shown in Figure 8 will be discussed along with compositional (Figure 9) and morphological (Figure 7) features for each Ag NPs/SiO_2_ sample, below.

Firstly, chromatic changes in the sample placed near the ventilation fan are discussed in comparison with those in the samples placed in ambient air and in the clean desiccator. The normalized LSPR band intensity of this sample decreased drastically compared to other samples as shown in Figure 8a. In addition, the LSPR band shifts quickly toward a longer wavelength as can be seen in Figure 8b. In this sample, the large worm-like aggregates due to particle coarsening were observed, and relatively large amounts of S and Cl atoms were detected. In the previous study [32], the Cl adsorption on clean Ag surfaces led to drastic morphological changes due to the formation of silver chloride (AgCl) islands. Similar situations would occur on the Ag NPs stored near the ventilation fan. In our case, the formation of AgCl caused the particle coarsening. Large amount of extrinsic impurities S and Cl as well as particle coarsening resulted in drastic changes in the LSPR intensity and wavelength. Thus, in an atmospheric environment containing S and Cl, the color of the Ag NPs turned bluish and then faded quickly as shown in Figure 2. Conversely, bluish coloring followed by a quick discoloring suggests significant amount of S and Cl in the environment. Of these elements, Cl would be mainly carried by dust such as particle matter (PM) because Cl was not detected in the samples in the clean and metal desiccators. In fact, ammonium salt, NH_4_Cl, is a common component of PM, and Qu et al., [33] investigated that deposition of NH_4_Cl leads to the atmospheric corrosion of zinc. The atmospheric environment of this sample is essentially the same with ambient air. It is probable that a one-way flow of air through the ventilation fan stimulates the adsorption of such extrinsic impurity elements on Ag NPs, resulting in the fastest discoloring. Thus, relatively quick monitoring of environment qualities can be achieved by one-way flow of the air using a ventilation fan.

Secondly, chromatic changes in the Ag NPs stored into the metal desiccator are discussed. For this sample, a remarkably blue-shift of the LSPR band was observed following the slightly red-shift. The color of this sample became reddish after 14 days passed, different from that of the other samples. The atomic ratio N/Ag for this sample increased to 0.66 at 14 days, remarkably higher than that for the other samples. As indicated by XPS and RBS, the Ag-CN and/or Ag-HCN compounds formed on the Ag NPs. It is probable that the origin of the chromatic change may be such compound formation, followed by surface roughening as revealed by AFM. More importantly, reddish coloring indicates the presence of CN and/or HCN inside the metal desiccator. Considering that the cyanide is used in stainless steel manufacturing [34], there is a possibility that it is released from the components of the metal desiccator.

Finally, the mechanism of a change in optical properties of Ag NPs/SiO_2_ is discussed briefly to understand the behavior shown in Figure 8. As described above, metallic Ag NPs exhibit strong LSPR in the visible region. The formation of compound, e.g., Ag_2_S, AgCl and AgCN, layers on Ag NP surfaces exposed to air reduces the volume of metallic Ag NPs. The reduction in volume of metallic Ag NPs enlarges the gap between adjacent NPs. The volume reduction and the gap widening result in the weakening of LSPR intensity (referred to as “effect 1”, hereafter) and the blue-shift of LSPR wavelength (effect 2) [35]. For a single metallic NP covered with a thin layer with a high refractive index, the LSPR position shifts toward longer wavelength (effect 3) together with enhancement in LSPR intensity (effect 4), depending on its refractive index as well as thickness [36]. In fact, McMahon et al. [23] calculated Mie scattering efficiency of a single Ag NP with various Ag_2_S thicknesses, and obtained results concerning the effect 1 and effect 3, i.e., reduced intensity and red-shift of LSPR with increasing Ag_2_S thickness. The calculated changes are identical to the LSPR changes observed for the sample placed near the ventilation fan. The LSPR changes observed for the sample stored in the metal desiccator were unique in the LSPR wavelength, i.e., blue-shift, which could be explained by the effect 1 and effect 2. In this sample, less influence of effect 3 would be due to the lower refractive index of AgCN (1.7 [37]) compared with those of AgCl (2.0 [38]) and Ag_2_S (2.2 [39]). The LSPR wavelength shifts as a function of refractive index were experimentally obtained by Sugawa et al. [40] for Ag NPs. It can be, therefore, considered that the refractive index of the overlayer formed on Ag NP surfaces determines the LSPR wavelength changes, red-shift or blue-shift.

Apart from the chromatic change, it should be noted that Br was detected on Ag NPs inside the metal desiccator, indicating that Ag NPs possess an ability to adsorb very toxic Br vapor. Ag NPs can, therefore, be used as adsorbent to detect trace amounts of toxic Br as well as cyanide because of their high reactivity.

## 5. Conclusions

The origin of chromatic changes is examined from the viewpoints of compositional and morphological changes in Ag NPs/SiO_2_ stored in various environments to develop a small and cheap device for monitoring of the cleanliness of the atmosphere. It was demonstrated that bluish coloring followed by quick discoloring of Ag NPs/SiO_2_ was induced by adsorption of a significant amount of S and Cl as well as particle coarsening due to the formation of AgCl, as was observed for the sample stored near the ventilation fan. Of these elements, Cl was probably carried by the particle matter containing Cl compounds, e.g., NH_4_Cl, considering that Cl was only detected in unenclosed samples. A remarkable blue-shift of the LSPR band, i.e., reddish coloring, was observed for the sample stored into the metal desiccator, resulting from the formation of Ag-CN and/or Ag-HCN compounds as well as surface roughening. The chromatic change into reddish color shows the inclusion of CN and/or HCN molecules inside the environment. Furthermore, the discoloration rate, defined as time required to become transparent, would be sensitive to the concentration of material to be adsorbed, meaning that the discoloration rate is a measure of the cleanliness of the air.

As expected from the present work, Ag NPs/SiO_2_ will be a promising material to check the quality of the environment. Long-term monitoring is more important than quick sensing for substances that accumulate in the human body and electronic materials, such as gaseous pollutants and PM. Our findings provide a pathway to develop a device that can easily check the cleanliness of the air by monitoring chromatic changes in Ag NPs/SiO_2_ with the naked eye. In addition, a fiber optic sensing system based on optical absorption changes, similar to the chemical gas sensor proposed by Chen et al. [41], can be developed using Ag NPs/SiO_2_, if quick sensing rather than monitoring is required.

## Figures and Tables

**Figure 1 nanomaterials-11-00701-f001:**
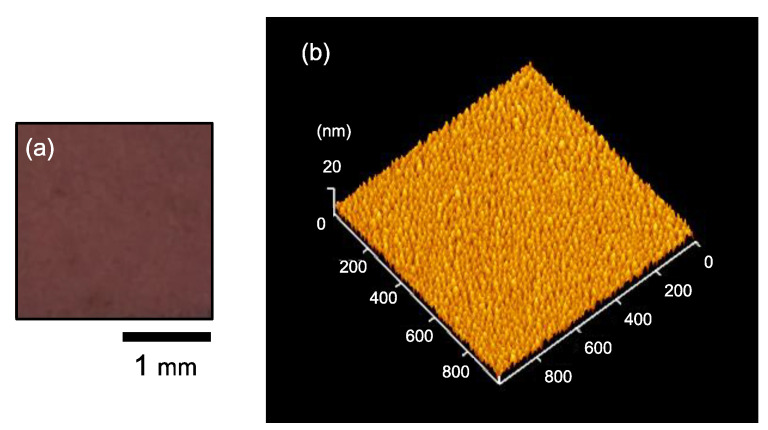
(**a**) A photograph of as-prepared Ag/SiO_2_; (**b**) and an atomic force microscopy (AFM) image of as-prepared Ag/Si sample that was aged in vacuum for 7 days.

**Figure 2 nanomaterials-11-00701-f002:**
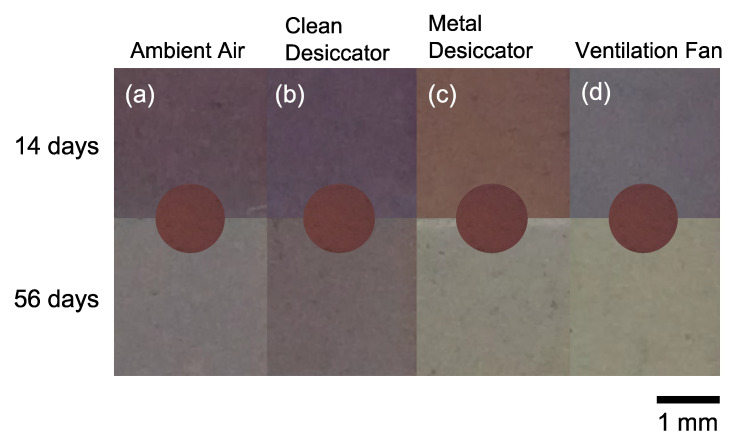
The photographs of Ag/SiO_2_ stored (**a**) in ambient air, (**b**) inside the clean desiccator, (**c**) inside the metal desiccator and (**d**) near the ventilation fan. Inset shows the photograph of as-prepared sample for comparison.

**Figure 3 nanomaterials-11-00701-f003:**
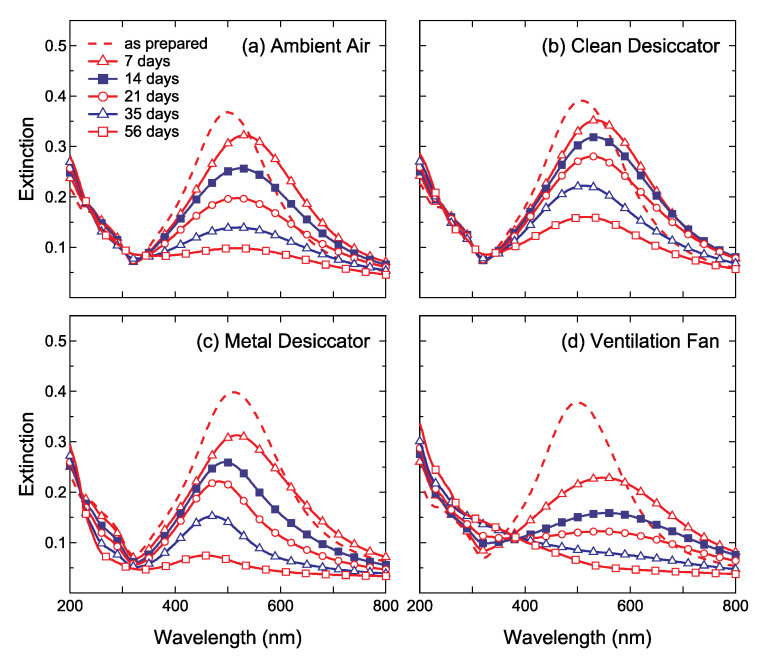
The optical extinction spectra of Ag/SiO_2_ stored (**a**) in ambient air, (**b**) inside the clean desiccator, (**c**) inside the metal desiccator and (**d**) near the ventilation fan. at room temperature for 7 days (open triangles), 14 days (filled squares), 21 days (open circles), 35 days (open triangles), 56 days (open squares). The spectrum of as-prepared sample (dashed lines) is also shown for comparison.

**Figure 4 nanomaterials-11-00701-f004:**
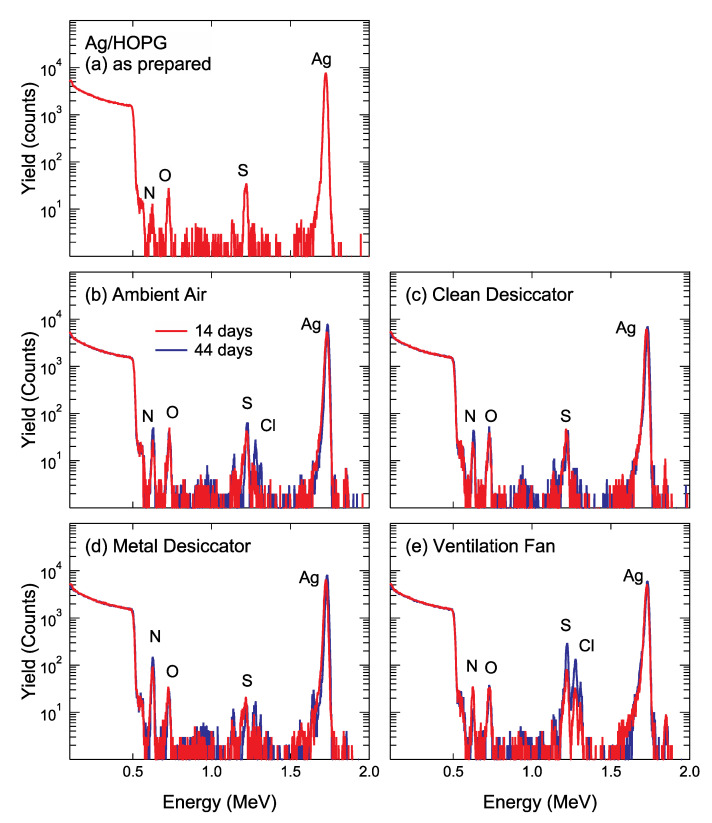
Backscattering spectra of (**a**) as prepared sample, Ag/highly oriented pyrolytic graphite (HOPG) stored (**b**) in ambient air, (**c**) inside the clean desiccator, (**d**) inside the metal desiccator and (**e**) near the ventilation fan for 14 days and 44 days.

**Figure 5 nanomaterials-11-00701-f005:**
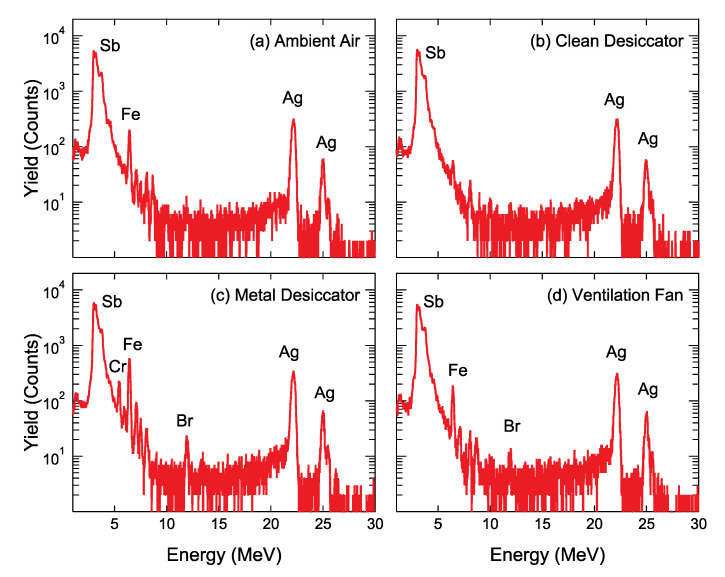
Particle-induced X-ray emission (PIXE) spectra of Ag/polyethylene naphthalate (PEN) stored (**a**) in ambient air, (**b**) inside the clean desiccator, (**c**) inside the metal desiccator and (**d**) near the ventilation fan for 28 days.

**Figure 6 nanomaterials-11-00701-f006:**
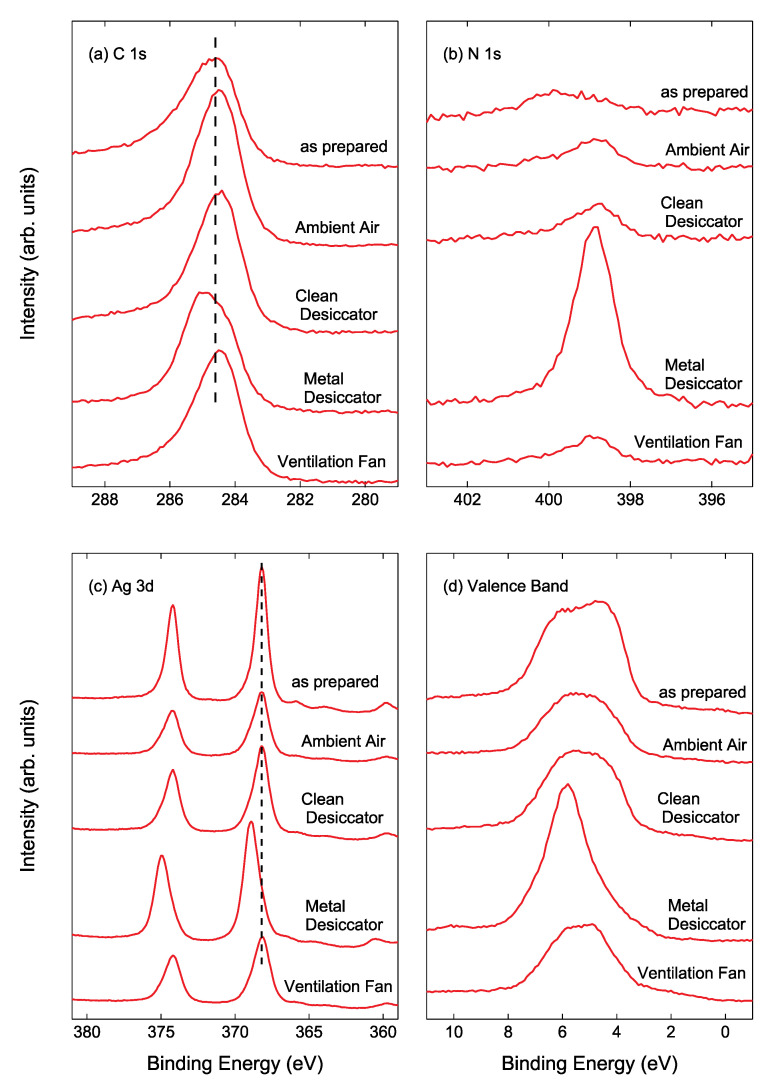
X-ray photoelectron spectroscopy (XPS) (**a**) C 1s, (**b**) N 1s, (**c**) Ag 3d and (**d**) valence band spectra of as-prepared Ag/HOPG sample, sample stored in ambient air, inside the clean desiccator, inside the metal desiccator and near the ventilation fan for 14 days.

**Figure 7 nanomaterials-11-00701-f007:**
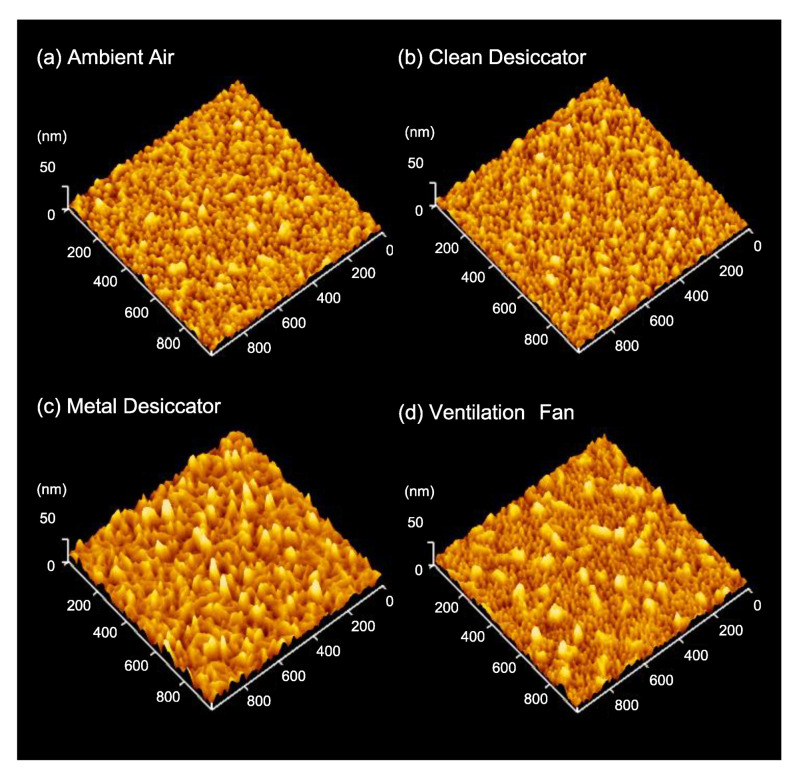
AFM images of Ag/Si stored (**a**) in ambient air, (**b**) inside the clean desiccator, (**c**) inside the metal desiccator and (**d**) near the ventilation fan for 14 days.

**Figure 8 nanomaterials-11-00701-f008:**
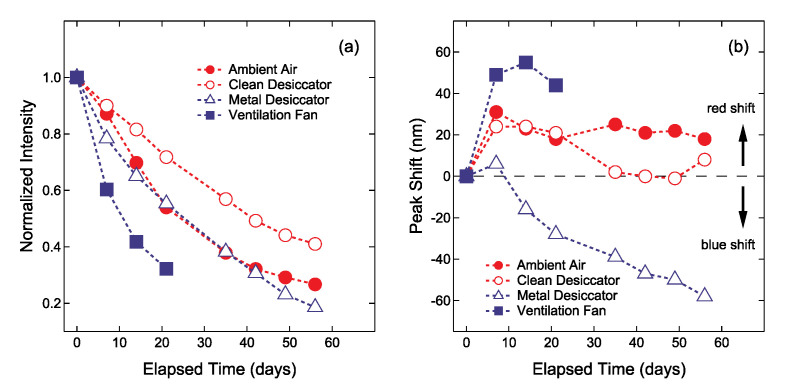
(**a**) Normalized extinction intensity and (**b**) peak shift of the localized surface plasmon resonance (LSPR) band of Ag NPs stored in ambient air (filled circles), inside the clean desiccator (open circles), inside the metal desiccator (open triangles) and near the ventilation fan (filled squares) plotted as a function of elapsed time.

**Figure 9 nanomaterials-11-00701-f009:**
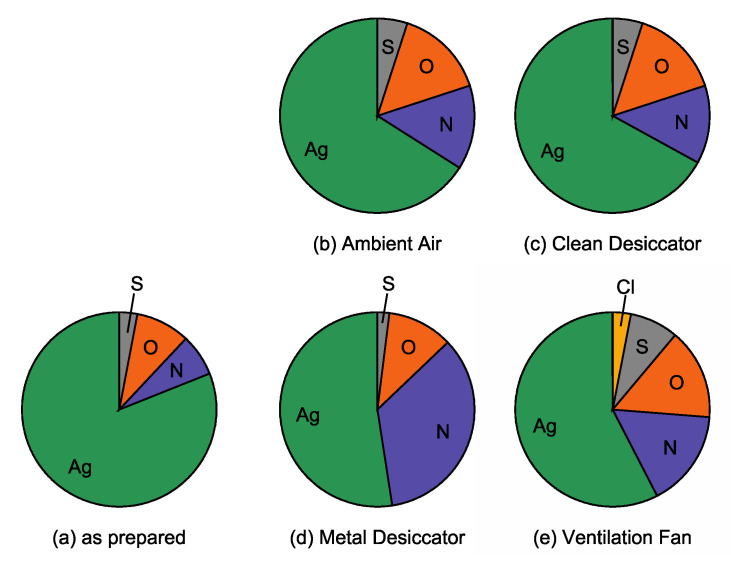
Pie charts showing the atomic ratios of impurity elements onto (**a**) as prepared sample, Ag/HOPG stored (**b**) in ambient air, (**c**) inside the clean desiccator, (**d**) inside the metal desiccator and (**e**) near the ventilation fan for 14 days determined by Rutherford backscattering spectrometry (RBS).

**Table 1 nanomaterials-11-00701-t001:** The atomic ratios of impurity elements M(*) to Ag determined by RBS.

Places	Atomic Ratios of M^(*)^ to Ag			
	N	O	S	Cl
As-prepared	0.08	0.11	0.04	−
Ambient air	0.21	0.23	0.08	−
Clean desiccator	0.20	0.22	0.07	−
Metal desiccator	0.66	0.20	0.04	−
Ventilation fan	0.28	0.27	0.14	0.06

(*) M = N, O, S, Cl.

## Data Availability

The data is available on reasonable request from the corresponding author.

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
