# Peer review of "Color Changes in Ag Nanoparticle Aggregates Placed in Various Environments: Their Application to Air Monitoring"

_nanomaterials, 2021, doi:10.3390/nano11030701_

Round 1

Reviewer 1 Report

In this manuscript, authors described a silver nanoparticle-based sensor for different air condition monitoring. I decided that the overall composition or contents are appropriate to be published in Nanomaterials. On the other hand, I decided that we could provide meaningful information to the readers only when the following matters are additionally supplemented.

(1) Detailed information on the product name and company name of the items required in this study (e.g., SiO2 plate) such as chemicals and wafers should be provided.

(2) A scale bar is required for each figure to provide the reader with the scale in Figure 1(a) and Figure 2.

(3) Two Ag peaks are shown in the PIXE spectra in Figure 5, and adding information on why two peaks occur is needed.

(4) I decided that the information on how this sensor can be applied should be presented with reference articles in Concluding Remarks.

Reviewer 2 Report

Koichi Ozaki and coauthors report on glass substrates modified with ag nanoparticles. these substrates show a color change over time. Furthermore, it was found that the color change was in dependence of the "quality" of the air. Especially the presence of chemical components like Cl and S led to a faster color change. The color change can be attributed to an impairment of the plasmonic properties of the ag nanoparticles, as found by UV/vis spectroscopy. However, the response mechanism of the nanoparticles is not fully elucidated, which is a weak point of the manuscript. It would be advisable for the authors to provide further experimental data to answer this open question. Overall, I find the work quite interesting, however, I cannot support its publication in the current state. I recommend detailed revisions and reconsideration after resubmission. comments below, in addition to the issues mentioned above, should help revise the manuscript and improve its significance and importance.  Finally, I must say that this paper fits well with the journal Nanomaterials, but does not meet the usual requirements in terms of presentation quality.

Comments/questions:

  1. Please discuss how this could be further developed to serve as reliable sensor for air quality.
  2. Figure 1a is missing a scale bar.
  3. Explain the aging step of the substrate, what is expected to happen and why it is necessary in more details.
  4. How was the “clean” desiccator cleaned?
  5. Figure 2 is missing scale bars.
  6. Oxidation/sulfidation of Ag at air is a commonly witnessed phenomenon. How could the formation of a coating of Ag2O or Ag2S be quantified experimentally?
  7. The term “SS desiccator” is not very comprehensive, I would recommend to avoid the abbreviation SS and simply use “steel” or “metal” instead (of course with a clear indication in the text that the metal is stainless steel).
  8. Were the sample stored in light or dark conditions? Light is known to strongly affect the stability of Ag nanoparticles. This is an important factor to take into account.
  9. I recommend to substitute the label “ventilation fan” to “laboratory air”.
  10. In Fig. 7d the surface seems to exhibit lager structures which make if differ from the others shown in Fig. 7abc. Is this correct? Please clarify.
  11. Minor typo (L.105): mode -> made.

Round 2

Reviewer 2 Report

The authors have revised their contribution and I basically agree with almost all the changes. The explanation of the response mechanism in lines 302-323 does not seem to me to be completely clarified, but the effects described are plausible. However, I must note that my comment #10 was misunderstood. For this reason I would like to ask again. In Figure 7d, the surface structures shown seems to exhibit different topographies which appear to be larger than those topographies shown in 7abc. Please clarify if this is correct and what changes in the surface topographies could indicate. Apart from these minor issues, I recommend publiction. 

Author Response

We sincerely thank  the reviewer for  valuable comments again. Also we must apologize for misunderstanding your question. As has been described in our manuscript at lines 238-239 and 271-276, the morphology shown in Fig. 7d was rather unique regarding particle coarsening as well as  worm-like features. That is our answer.